# Visceral Adipose Tissue Molecular Networks and Regulatory microRNA in Pediatric Obesity: An In Silico Approach

**DOI:** 10.3390/ijms231911036

**Published:** 2022-09-20

**Authors:** Dipayan Roy, Anupama Modi, Ritwik Ghosh, Raghumoy Ghosh, Julián Benito-León

**Affiliations:** 1Department of Biochemistry, All India Institute of Medical Sciences (AIIMS), Jodhpur 342005, Rajasthan, India; 2Indian Institute of Technology (IIT), Madras 600036, Tamil Nadu, India; 3School of Humanities, Indira Gandhi National Open University (IGNOU), New Delhi 110044, Delhi, India; 4Department of General Medicine, Burdwan Medical College & Hospital, Burdwan 713104, West Bengal, India; 5Lee Kong Chian School of Medicine, Nanyang Technological University (NTU), Singapore 636921, Singapore; 6Department of Neurology, University Hospital “12 de Octubre”, Av. De Córdoba, s/n, 28041 Madrid, Spain; 7Centro de Investigación Biomédica en Red sobre Enfermedades Neurodegenerativas (CIBERNED), Av. De Córdoba, s/n, 28041 Madrid, Spain; 8Department of Medicine, Universidad Complutense, Pl. de Ramón y Cajal, s/n, 28040 Madrid, Spain

**Keywords:** childhood obesity, in silico, microRNA, obesity, visceral adipose tissue

## Abstract

Childhood obesity carries an increased risk of metabolic complications, sleep disturbances, and cancer. Visceral adiposity is independently associated with inflammation and insulin resistance in obese children. However, the underlying pathogenic mechanisms are still unclear. We aimed to detect the gene expression pattern and its regulatory network in the visceral adipose tissue of obese pediatric individuals. Using differentially-expressed genes (DEGs) identified from two publicly available datasets, GSE9624 and GSE88837, we performed functional enrichment, protein–protein interaction, and network analyses to identify pathways, targeting transcription factors (TFs), microRNA (miRNA), and regulatory networks. There were 184 overlapping DEGs with six significant clusters and 19 candidate hub genes. Furthermore, 24 TFs targeted these hub genes. The genes were regulated by miR-16-5p, miR-124-3p, miR-103a-3p, and miR-107, the top miRNA, according to a maximum number of miRNA–mRNA interaction pairs. The miRNA were significantly enriched in several pathways, including lipid metabolism, immune response, vascular inflammation, and brain development, and were associated with prediabetes, diabetic nephropathy, depression, solid tumors, and multiple sclerosis. The genes and miRNA detected in this study involve pathways and diseases related to obesity and obesity-associated complications. The results emphasize the importance of the TGF-β signaling pathway and its regulatory molecules, the immune system, and the adipocytic apoptotic pathway in pediatric obesity. The networks associated with this condition and the molecular mechanisms through which the potential regulators contribute to pathogenesis are open to investigation.

## 1. Introduction

Pediatric obesity is associated with an increased risk of metabolic complications, including diabetes mellitus, metabolic syndrome, cardiovascular disease, sleep disturbances, and certain malignancies later in life [1,2,3]. In the past few decades, the prevalence of obesity in the pediatric age group has sharply increased in developing and developed countries [1], making it a global health burden.

Obesity-related insulin resistance (IR) develops due to inflammation caused by excessive fat accumulation and abnormalities in lipid metabolism. Adipose tissue (AT) is a central node in regulating our metabolic activities and energy homeostasis. The visceral adipose tissue (VAT) component has a higher propensity for inflammation; its strong association with IR in contrast to subcutaneous fat depots is well established. The ongoing lipotoxicity, endoplasmic reticular stress, oxidative stress, recruitment of AT macrophages, and subsequent secretion of pro-inflammatory cytokines in an obese state are said to impair the insulin signaling pathway in the liver and skeletal muscle leading to IR [4,5]. However, the pathogenic molecular alterations in VAT leading to IR are not entirely known. Recent evidence suggests that adipocyte gene expression and the transcriptional regulation of gene products have a central role in IR in long-term obesity.

Visceral adiposity is independently associated with IR and inflammation in obese adolescents, highlighting its potential role in obesity-related chronic disease [6]. Moreover, a differential transcriptional response to hyperinsulinemia in obese subjects compared to non-obese individuals was found to be independent of insulin sensitivity [7], implying that VAT is the potential culprit. We can, therefore, hypothesize that the differentially-expressed genes in VAT and their transcriptional regulators are involved in altering the roles of key proteins in the insulin signaling pathway and glucose homeostasis in obese conditions.

Furthermore, microRNAs (miRNAs), well-known for their role in diabetes and its complications [8,9], have shown promise as markers of endothelial dysfunction, type 2 diabetes, metabolic syndrome, and IR in children [10]. The miRNAs produced by AT depots can affect the metabolism of distant tissues because they are easily transported into the bloodstream. Strycharz and colleagues [11] identified several differentially-expressed miRNAs in the VAT of pre-diabetic and diabetic females, stressing that these expression changes could be associated with the low-grade chronic inflammation and oxidative stress in VAT in hyperglycemia. Both transcription factors (TFs) and miRNA are known to regulate the genetic network associated with insulin secretion and β-cell proliferation in the context of obesity. A previous study [12] reported that AT macrophages modulate obesity-related β-cell adaptations through the secretion of miRNA-containing (miR-155) extracellular vesicles. Recently, several miRNAs, such as miR-320a, miR-142-3p, and the let-7 family, were identified as potential circulating biomarkers of IR in pre-adolescent children [13].

Our study aimed to discover any existing pattern of gene expression and its regulatory network in VAT of the obese pediatric population. We identified differentially-expressed genes (DEGs) for VAT by analyzing the mRNA expression profiles of two different microarray datasets. Subsequently, Gene Ontology (GO) and functional enrichment were carried out to reveal the pathways and mechanisms involved in childhood obesity. The protein–protein interaction (PPI) network was constructed, and hub genes were identified. Finally, we analyzed the candidate hub genes for target miRNA and TFs. The mechanisms, the hub genes, their co-expressed regulators, and the potentially involved pathways can reveal useful information in understanding the mechanism behind childhood obesity.

## 2. Results

### 2.1. Identification of DEGs between Obese and Lean Individuals

A schematic diagram depicting the study design is shown in Figure 1.

For the chosen datasets, the number of upregulated DEGs was 1182 (GSE9624, obese vs. lean) and 378 (GSE88837, obese vs. lean), whereas the number of downregulated DEGs was 1226 (GSE9624, obese vs. lean) and 514 (GSE88837, obese vs. lean), respectively. Figure 2A,B depicts the significant DEGs in the two datasets. Venn diagrams were then constructed using the Venn diagram tool available at http://bioinformatics.psb.ugent.be/webtools/Venn/ (accessed on 11 July 2022) to identify the overlapping DEGs between the two groups. Eighty-one upregulated (Figure 2C) and 103 downregulated DEGs (Figure 2D) were identified as overlapping between the two datasets, which were selected for further analysis.

### 2.2. Functional Enrichment Analysis of DEGs

The top 20 significant GO enrichments for the overlapping DEGs are displayed in Figure 3A,B. The upregulated genes were enriched in response to hormones, enzyme-linked receptor protein signaling pathway, positive regulation of the apoptotic process, response to estradiol, hemopoiesis, regulation of growth, cellular response to fatty acid, positive regulation of protein kinase B signaling, regulation of fat cell differentiation, and regulation of cell adhesion and were involved in the TGF-β signaling pathway (Figure 3A). The downregulated genes are strongly enriched in Complement Cascade, Cell Cycle-mitotic, regulating cholesterol metabolic process, inflammatory response, and connective tissue development. These genes are also involved in the IL-18 signaling pathway, trans-sulfuration, one-carbon metabolism, and ferroptosis (Figure 3B).

Furthermore, the disease–gene interaction from DisGeNet revealed that the upregulated genes were involved in vascular inflammations, unipolar depression, weight gain, kidney failure, and adolescent idiopathic scoliosis (Figure 3C). In contrast, the downregulated genes were significantly associated with meningioma, acute myocardial infarction, diseases of the capillaries, Behçet syndrome, and Fatty Liver disease (Figure 3D). The functional enrichment in terms of Biological Process, Molecular Function, and Cellular Components is given in Appendix A.

### 2.3. PPI Network Construction and Significant Modules

The PPI network on the Search Tool for the Retrieval of Interacting Genes and proteins (STRING) platform was significantly enriched (*p* < 0.001). The network was imported and visualized on Cytoscape (Figure 4).

We found 44 upregulated and 71 downregulated overlapping genes in this network with 552 paired interactions (Appendix A).

The network analysis through the MCODE plug-in was carried out with a Network Score Degree Cutoff: 2, and Cluster parameters—Node Density Cutoff: 0.1, Node Score Cutoff: 0.2, K-Core: 2, and Maximum Depth: 100. The analysis revealed six significant clusters. Cluster 1, with the highest score of 13.077, had 14 nodes and 170 edges and included Kinesin Family Member 20A (KIF20A), Aurora Kinase A (AURKA), Hyaluronan Mediated Motility Receptor (HMMR), dual specificity protein kinase TTK, DNA Topoisomerase IIα (TOP2A), Centromere Protein N (CENPN), Denticleless protein homolog (DTL), Lymphoid-specific helicase (HELLS), G2/mitotic specific Cyclin-B2 (CCNB2), GINS complex subunit 2 (GINS2), Kinetochore protein SPC24, Kinesin Family Member 4A (KIF4A), Centromere Protein F (CENPF), Thymidylate synthase (TYMS) (Figure 5A). The details of each module are given in Appendix A.

Cluster 2, having six nodes, 28 edges, and a score of 5.600, included six genes: Leptin (LEP), Sterol Regulatory Element Binding Transcription Factor 1 (SREBF1), ATP Citrate Lyase (ACLY), Pyruvate Dehydrogenase Kinase 4 (PDK4), Stearoyl-CoA Desaturase (SCD), and Diacylglycerol O-acyltransferase 2 (DGAT2) (Figure 5B). Cluster 3, having 11 nodes, 38 edges, and score: 3.800, had eleven genes: namely, Complement Receptor type 1 (CR1), Haptoglobin (HP), Kininogen 1 (KNG1), Tissue inhibitor of metalloproteinases 1 (TIMP1), Apolipoprotein E (APOE), Type-I Collagen alpha-1 (COL1A1), Complement Factor I (CFI), Versican (VCAN), Complement Factor B (CFB), Matrix Metalloproteinase 14 (MMP14), and Type-V Collagen alpha-1 (COL5A1) (Figure 5C).

Clusters 4, 5, and 6 (each having three nodes, six edges, and a score of 3.000) had three genes each. They were α1 antitrypsin (SERPINA1), Complement C4-A (C4A), and Complement C4-B (C4B) (Cluster 4) (Figure 5D); JunD proto-oncogene (JUND), MAF bZIP transcription factor F (MAFF), and NADPH quinone oxidoreductase 1 (NQO1) (Cluster 5) (Figure 5E); and serine hydroxymethyltransferase 1 (SHMT1), Betaine-Homocysteine S-methyltransferase 2 (BHMT2), and Cystathionine β synthase-like (CBSL) (Cluster 6) (Figure 5F), respectively.

### 2.4. Hub Gene Identification and TF–Gene Interactions

The topmost hub genes were identified by the cytoHubba plug-in using twelve topological algorithms, and the genes detected in at least three techniques were considered candidate hub genes. A total of 19 such genes were identified, as shown in Table 1.

The TF–gene interaction network revealed 90 hub gene–TF pairs (Figure 6, Appendix A). These genes were involved in clusters 1, 2, 3, and 6. There were 210 and 79 targeting TFs between these hub genes, respectively, according to Encyclopedia of DNA elements (ENCODE) and Transcriptional Regulatory Relationships Unravelled by Sentence-based Text Mining (TTRUST)—two databases incorporated in miRNet for uncovering the molecular basis of TF–binding. The Venn diagram tool was used to identify the 24 common targeting TFs between these two databases.

### 2.5. MicroRNA and Hub Gene Network

Hub gene-targeting miRNA were predicted using miRNet based on the correlation analysis between the hub genes and miRNAs, using a degree cut-off of 2.0. The regulatory network predicting the miRNA-hub gene interaction is shown in Table 2 and Figure 7. In the network, there were 19 genes and 115 miRNA. The top 20 targeting miRNA were selected from the interaction table, with 228 mRNA–miRNA pairs (Appendix A).

The top five targeting miRNA according to a maximum number of interaction pairs were miR-34a-5p, miR-16-5p, miR-124-3p, miR-103a-3p, and miR-107. The functional analysis yielded eight items for cell specificity, four for the clusters, 262 for disease, four for miRNA family, 56 functions, and four for tissue specificity. All items were sorted according to the statistical significance per the false discovery rate (FDR), and relevant functions and disease categories were visualized. The selected functional annotations included glucose and lipid metabolism, brain development, immune response, vascular inflammation, smooth muscle cell proliferation, hormone-mediated signaling pathway, and T-cell differentiation (Figure 8A). The significant disease categories included inflammatory bowel disease, type 2 diabetes mellitus, diabetic nephropathy and retinopathy, solid childhood tumor, prediabetes, depression, polycystic ovarian syndrome, and multiple sclerosis (Figure 8B).

## 3. Discussion

Childhood obesity is a prevalent global health issue that needs our immediate attention. A dysfunctional AT is believed to be responsible for the development of obesity-associated metabolic disorders. Several studies have used microarray data profiling to elucidate the pathogenic mechanisms of pediatric obesity. Aguilera and colleagues identified a distinct gene expression pattern in VAT in obese children [14]. The validated genes in their study were involved in lipid and amino acid metabolism, oxidative stress, adipogenesis, and inflammation. Findings from another experiment suggested a dysfunctional PI3K/Akt signaling pathway in the VAT of obese children, which may be driven by changes in DNA methylation [15]. The present study identified 184 overlaps and differentially-expressed RNAs from both datasets. Following functional enrichment analysis, the upregulated DEGs were involved in hormone response, cellular response to fatty acids, regulation of adipocyte differentiation, positive regulation of Akt signaling, and regulation of cell adhesion, which are in line with the previous findings. The disease–gene interactions further revealed several conditions, some of which are known to be associated with obesity, such as heart disease, fatty liver disease [16], and nephropathy.

Other significant biological processes were the TGF-β receptor signaling pathway and the integrin-linked kinase (ILK) signaling pathway. The former is well-researched in obesity and takes part in multiple pathophysiological roles in cardiometabolic diseases [17]. An imbalance of TGF-β and interleukin-10 (IL-10) levels in neutrophils aggravates the inflammatory cytokine expression in childhood obesity [18]. It is already known that VAT exosomes can induce TGF-β pathway dysregulation in hepatocytes in vitro, indicative of a possible role in the pathogenesis of non-alcoholic fatty liver disease (NAFLD) [19]. Recently, a distinctive NAFLD-associated transcriptomic signature, which included a highly expressed TGF-β1, has been reported in the white AT of severely obese females [16]. Besides vascular inflammation, weight gain, kidney failure, and liver and heart diseases, the disease–gene interaction also revealed unipolar depression, various cancers, and scoliosis, all known risks for obese children [3,20,21]. Several cancer-related TFs such as JUNB, JUND, Forkhead box protein M1 (FOXM1), and ETS1 were part of the TF–mRNA regulatory network. These TFs all augment the TGF-β signaling pathway [22,23,24]. Studies on obese mice models have identified JUND as a key metabolic regulator of lipid metabolism and obesity-induced cardiomyopathy [23]. FOXM1 is upregulated in obesity and helps in β-cell proliferation as a compensatory mechanism in IR [25]. SMAD4, one of the targeting TFs in our study, is a central mediator in the TGF-β pathway. The latter is involved with and modulated by miR-124 (found to target 15 out of 19 candidate hub genes in the current study) through a feedback loop that includes SMAD4 [26]. The ILK pathway has a cause–effect relationship with muscle IR in obese mice [27].

The downregulated genes were involved in the complement cascade, neutrophil degranulation, apoptosis-related network, and IL-18 signaling pathway. The delayed resolution of acute inflammation in obesity due to the deficiency of an effective apoptotic adipocyte removal process can contribute to the underlying inflammatory process, immune system dysfunction, and IR [28]. M2 macrophages can help efficiently clear these apoptotic cells to reduce the burden of inflammation and long-term complications. Adiposity-related inflammatory factors also form inflammasome complexes, which activate and release pro-inflammatory cytokines, including IL-18, one of the crucial cytokines involved in the development of IR [29]. Recently, the role of the complement system has also emerged in obesity-associated metabolic disorders and adipocyte inflammation. Decreased levels of the lectin pathway components reduce the clearance of apoptotic adipocytes and increase AT inflammation [30]. This is supported by the complement component 4a deficiency in our analysis.

Among the hub genes, APOE is significantly reduced in the peripheral blood of obese children [31]. Two recent studies highlight the role of SREBF1 in childhood obesity [32,33]. Kochmanski et al. [32] studied DNA methylation patterns in neonatal blood spots and found both LEP and SREBF1 to be associated with growth and adiposity. Notably, epigenetic-based pharmacological strategies are being explored in VAT to ameliorate obesity-related comorbidities [34]. TIMP1 is a negative regulator of adipogenesis. It is increased in the AT of obese mice models [35]. HMMR is another regulatory molecule, the downregulation of which induces adipogenesis in vitro [36]. Signal transducer and activator of transcription 1 (STAT1) is a TF involved in IR and cancer; STAT1 regulates cellular proliferation, inflammation, and angiogenesis and mediates between reactive oxygen species (ROS)-induced damage, attenuation of macrophage differentiation and endothelial dysfunction in diabetic complications [37]. CEBPB and CEBPA (CCAAT/enhancer-binding protein-beta and alpha) are known TFs in the adipogenic differentiation cascade. Previous studies have elucidated their roles in AT’s developmental stage-specific transcriptional networks [38]. Several hub genes, such as TOP2A, CENPF, TYMS, AURKA, KIF4A, and KIF20A, are related to the cell cycle and DNA replication, implicating the close relationship between obesity and cancer [39,40]. AURKA is a prognostic marker in obese patients with early breast cancer [40]. Loss of AURKA in the intestinal epithelia causes gut microbiota dysbiosis and higher levels of propionate, leading to Akt activation, which in turn promotes obesity [41].

Obesity-associated or AT-derived miRNA has potential as biomarkers for managing and preventing obesity and as promising therapeutic targets. This study identified five miRNAs that targeted the network’s maximum number of hub genes. A recent in silico analysis determined miR-124-3p to be a key regulatory molecule in the pathogenesis of type 2 diabetes mellitus [42]. This miRNA targets the immune status of individuals through interacting with obesity-related immune cytokines [43]. Moreover, miRNAs such as miR-155-5p, miR-1-3p, and let-7b-5p—among the targeting miRNAs found in this study—are involved in the PI3k/Akt pathway, endocrine resistance, and advanced glycation end products/receptor for advanced glycation end products (AGE/RAGE) signaling pathway [42]. A comprehensive transcriptomic analysis involving type 2 diabetes mellitus patients further revealed that miR-124-3p and miR-16-5p affect the expression of genes such as CREB1, SP1, (both TFs in our analysis) SREBF1, and JUN (hub genes) [44].

Circulating levels of miR-16-5p have predictive value as a biomarker of gestational diabetes mellitus in obese pregnant women [45]. Increased miR-34 in AT exacerbates the inflammatory process by suppressing Kruppel-like factor 4 (KLF4), thereby increasing the accumulation of proinflammatory M1 macrophages [46]. MiR-103a-3p and miR-107 are known to act in the insulin signaling pathway [47]. Zhang and colleagues studied the effects of miR-103/107 on preadipocyte apoptosis. They showed that these two miRNAs promote endoplasmic reticulum-mediated apoptosis through the Wnt3a/β-catenin pathway [48], suggesting that activating these miRNAs could potentially serve as novel therapies for treating obesity and metabolic syndrome-related diseases.

Interestingly, inflammatory bowel diseases (IBD) and multiple sclerosis were significant disease annotations in the miRNA enrichment analysis. Recently, several lines of evidence have come up to suggest a link between visceral adiposity and IBD. The links involve a chronic inflammatory state, alteration in the gut microbiome, and diet. The adipokines released from the AT can be proinflammatory in immune-mediated disorders. Visceral adiposity and obesity have an impact on several IBD-related outcomes, such as response to therapy and quality of life [49]. A survey of the existing literature also shows a link between young overweight or obese individuals and the occurrence of multiple sclerosis, which was significant for girls [50].

This study has several limitations. The total sample size was small and only included individuals of specific ethnicities. Larger cohort studies are needed for the verification of our findings. The miRNA expression in obesity may also be gender-dependent, as shown in previous studies on different ethnicities [10,51]. This needs to be explored further in other populations. Finally, although we have identified the potential regulatory genes, TF, and miRNA, and the networks and pathways associated with childhood obesity, the molecular mechanisms of these potential regulators are still open to laboratory investigation in vitro or in vivo.

## 4. Materials and Methods

### 4.1. Microarray Datasets and Screening of DEGs

The expression profiling datasets GSE9624 [14] and GSE88837 [15] were obtained from the Gene Expression Omnibus (GEO) Datasets on the NCBI website (https://www.ncbi.nlm.nih.gov/gds) (accessed on 11 July 2022), organized according to *Homo sapiens*. The following inclusion criteria were set while selecting the samples for this study: VAT or omental AT from obese children or adolescents (2–19 years), with lean individuals used as control samples. GSE9624 had 11 samples (five obese and six normal-weight children), and GSE88837 had 29 samples (14 obese and 15 lean).

Both datasets were based on GPL570 [HG-U133_Plus_2] Affymetrix Human Genome U133 Plus 2.0 Array. The available data were processed using the interactive web tool GEO2R. The DEGs of our interest were segregated under the following threshold: *p*-value < 0.05 and |log_2_ (fold change)| ≥ 1.

### 4.2. DEG Functional Enrichment and Disease–Gene Interactions

Gene Ontology (GO) and pathway analysis for the overlapping DEG were analyzed on Metascape [52], an online tool for gene annotation. The genetic underpinning of the related diseases was determined through DisGeNet, an integration platform of curated databases within Metascape. Terms with an adjusted *p*-value < 0.05, a minimum count of 3, and an enrichment factor > 1.5 are collected and grouped into clusters based on their membership similarities.

### 4.3. PPI Network and Hub Gene Identification

PPI network of overlapping DEGs was constructed using the STRING database (https://string-db.org/) (accessed on 11 July 2022), which analyzes the functional interaction between proteins. To explore the regulatory mechanisms, interactions with the confidence of a combined score > 0.400 were retained and imported to Cytoscape (version 3.9.1, Cytoscape team, Institute for Systems Biology, Seattle, WA, USA) for visualization [53]. The molecular complex detection (MCODE) plug-in in Cytoscape was used for selecting the top clustering modules with the default settings. The cytoHubba app (Cytoscape 3.9.1, Cytoscape team, Institute for Systems Biology, Seattle, WA, USA) [54] was used to identify hub genes with the twelve topological methods. The top 10 genes were detected for each technique, and the genes present in at least three ways were considered hub genes.

### 4.4. Hub-Gene Targeting TF, miRNA Network, and Functional Enrichment

Hub gene-targeting miRNAs were predicted using miRNet 2.0 [55], an integrated platform for miRNA-centric network visual analytics. It integrates data from 14 different miRNA databases. Additionally, the hub gene-targeting TFs were analyzed as per two other platforms included in miRNet- ENCODE and TRRUST. The common transcription factors identified in both databases were used for the hub gene–TF interaction network visualized on Cytoscape.

TAM 2.0 (http://www.lirmed.com/tam2/) (accessed on 18 July 2022) is a miRNA set enrichment analysis tool for mining the functional and disease implication behind miRNAs of interest. The miRNAs are grouped into six categories according to family, cluster, condition, function, TF, and tissue specificity. The top five gene-targeting miRNAs in our analysis, chosen based on the number of interaction pairs, were used for enrichment analysis. By default, the overrepresentation option was chosen. We selected the “Up and down” option to analyze the up/downregulated miRNA sets.

## 5. Conclusions

VAT is linked to the development of obesity and obesity-related complications. However, the exact mechanism underlying the interconnection of VAT in obesity and related metabolic complications is still unclear. We identified 184 VAT-specific DEGs in the pediatric obese population from the selected datasets. A total of 19 candidate hub genes were selected from there to analyze targeting TF and miRNA. The miRNA identified in this study involves pathways and diseases related to obesity and associated complications. In the future, there is scope to explore these molecular pathways in larger cohorts and develop novel, miRNA-based therapeutics for obesity and metabolic diseases.

## Figures and Tables

**Figure 1 ijms-23-11036-f001:**
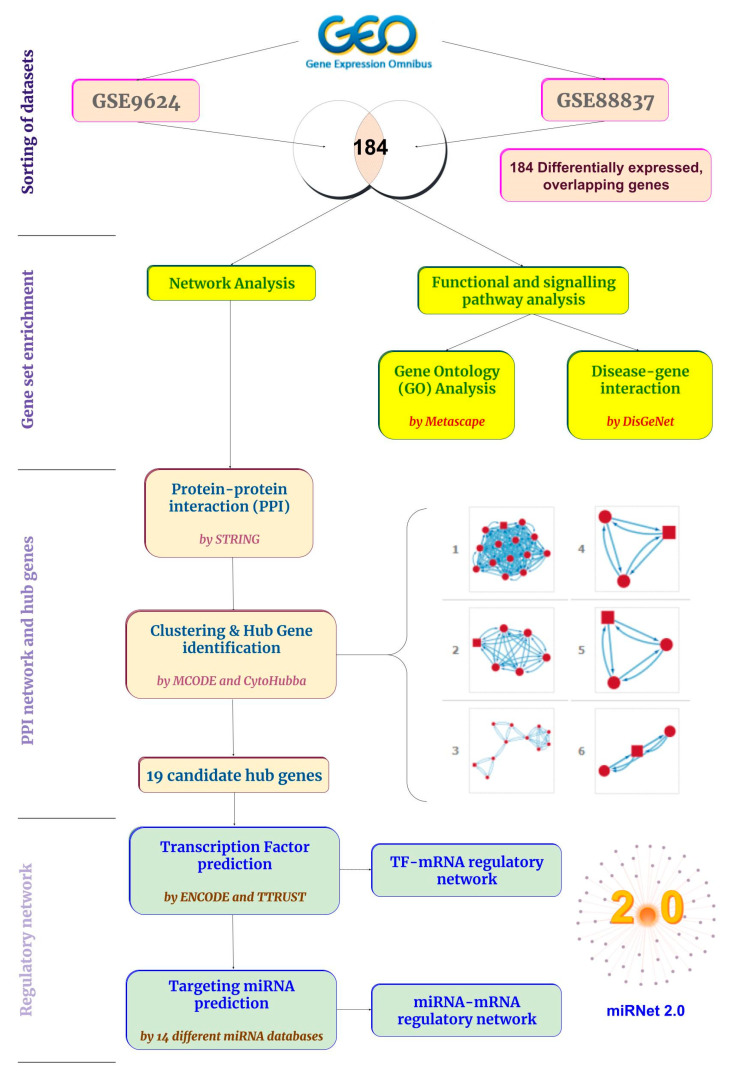
The schematic flow of the study.

**Figure 2 ijms-23-11036-f002:**
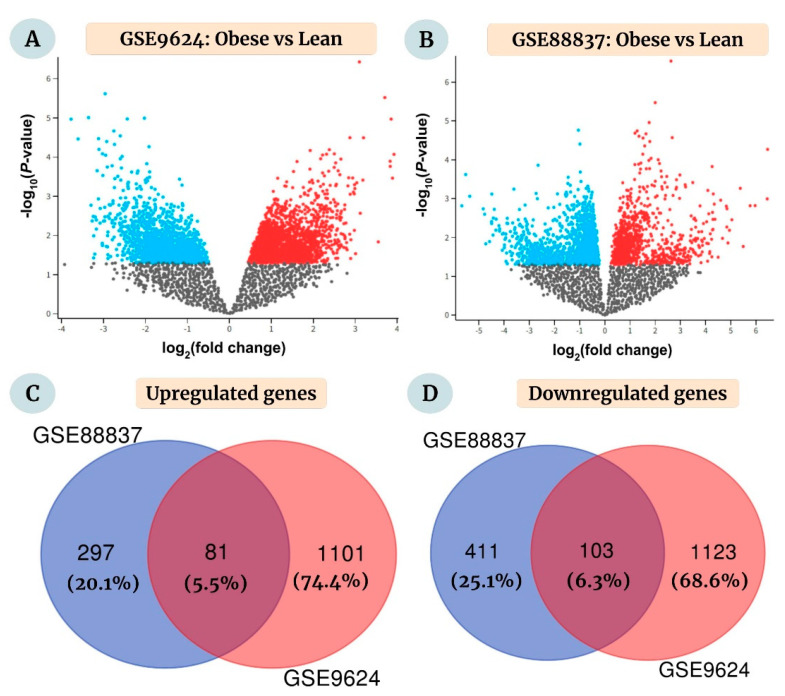
Volcano plots depicting the analysis of obesity-induced differentially-expressed genes (DEGs) in visceral or omental adipose tissue comparing obese and lean pediatric subjects (aged 2-19 years) in the datasets (**A**) GSE9624 and (**B**) GSE88837, respectively. The publicly available Gene Expression Omnibus (GEO) datasets, downloaded from the NCBI website, together comprised 40 samples from 19 obese and 21 lean children or adolescents. DEGs were obtained using the GEO2R online interactive tool (that uses the GEOquery, Limma, and umap R packages) with the cut-off *p*-value < 0.05. The downregulated and upregulated genes are depicted in blue and red, respectively, for both datasets. (**C**,**D**) The DEGs for the individual datasets were further segregated using the threshold |log_2_ (fold change)| ≥ 1. Venn diagrams indicate the overlap between (**C**) upregulated and (**D**) downregulated DEGs, respectively. These 184 common DEGs (81 upregulated and 103 downregulated) were selected for further analysis.

**Figure 3 ijms-23-11036-f003:**
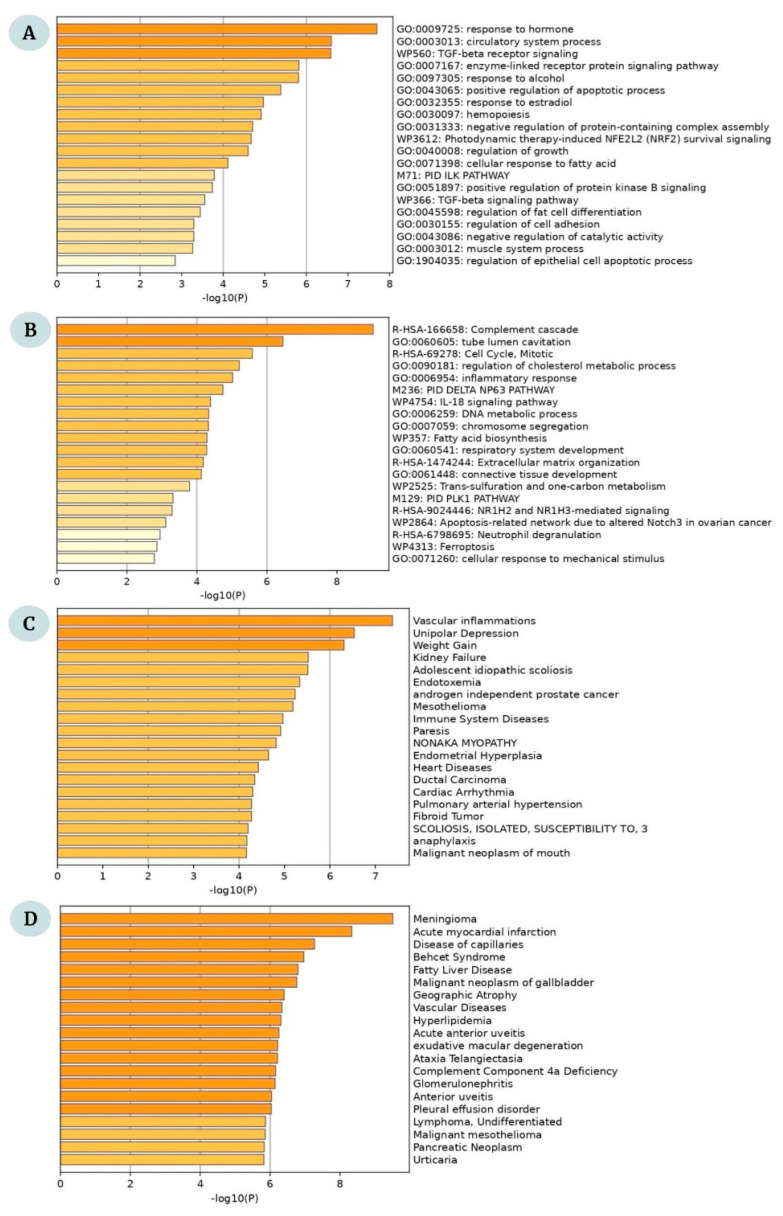
The GO enrichment analysis of the overlapping DEGs between obese and lean children. Top 20 significant enrichment terms for (**A**) upregulated and (**B**) downregulated DEGs. (**C**,**D**) The top 20 disease-gene interactions for the overlapping genes between obese and lean children.

**Figure 4 ijms-23-11036-f004:**
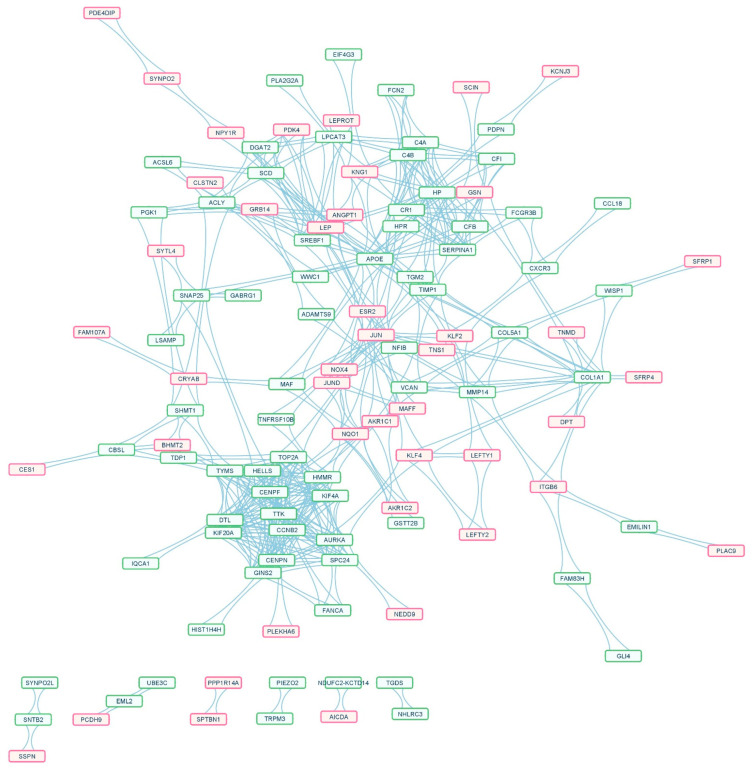
The protein–protein interaction (PPI) network of overlapping DEGs between obese and lean children. The red-purple border indicates the upregulated genes, and the green border shows the downregulated overlapping genes. There were 44 upregulated and 71 downregulated overlapping genes in the PPI network with 552 paired interactions.

**Figure 5 ijms-23-11036-f005:**
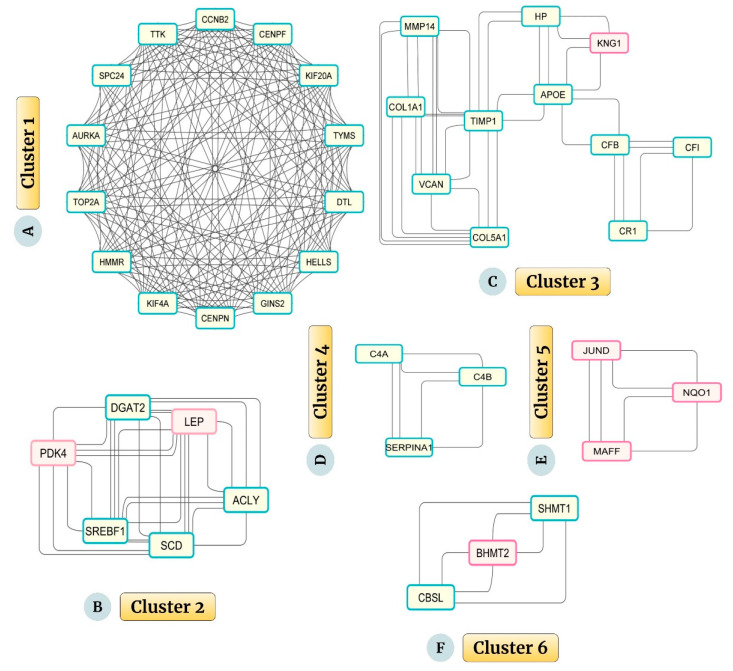
(**A**–**F**) The top six significant clusters were extracted from the PPI network through the MCODE plug-in. The red-purple border indicates the upregulated genes, and the green wall shows the downregulated overlapping genes. (**A**) Cluster 1 (KIF20A, AURKA, HMMR, TTK, TOP2A, CENPN, DTL, HELLS, CCNB2, GINS2, SPC24, KIF4A, CENPF, TYMS): 14 nodes, 170 edges, and score: 13.077, (**B**) Cluster 2 (LEP, SREBF1, ACLY, PDK4, SCD, DGAT2): 6 nodes, 28 edges, and score: 5.600, (**C**) Cluster 3 (CR1, HP, KNG1, TIMP1, APOE, COL1A1, CFI, VCAN, CFB, MMP14, COL5A1): 11 nodes, 38 edges, and score: 3.800, (**D**) Cluster 4 (SERPINA1, C4A, C4B): 3 nodes, 6 edges, and score: 3.000, (**E**) Cluster 5 (JUND, MAFF, NQO1): 3 nodes, 6 edges, and score: 3.000, (**F**) Cluster 6 (SHMT1, BHMT2, CBSL): 3 nodes, 6 edges, and score: 3.000.

**Figure 6 ijms-23-11036-f006:**
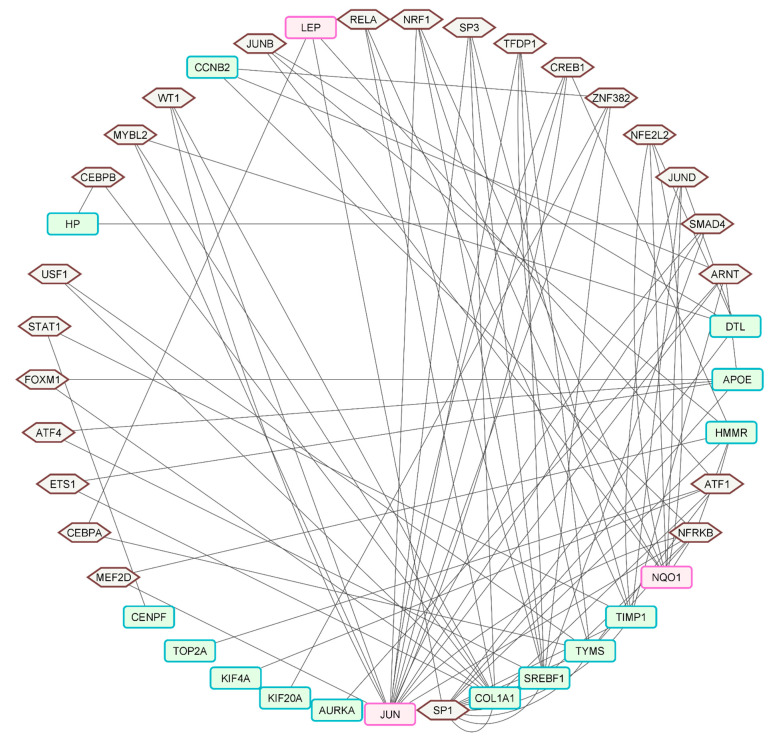
The hub gene-transcription factor (TF) network. The hexagons with brown outlines indicate the TFs, and the round, rectangular nodes are the hub genes, those upregulated marked with the red-purple border and those downregulated with the green border. The network had 17 hub genes and 24 TFs, with 90 hub gene–TF interaction pairs.

**Figure 7 ijms-23-11036-f007:**
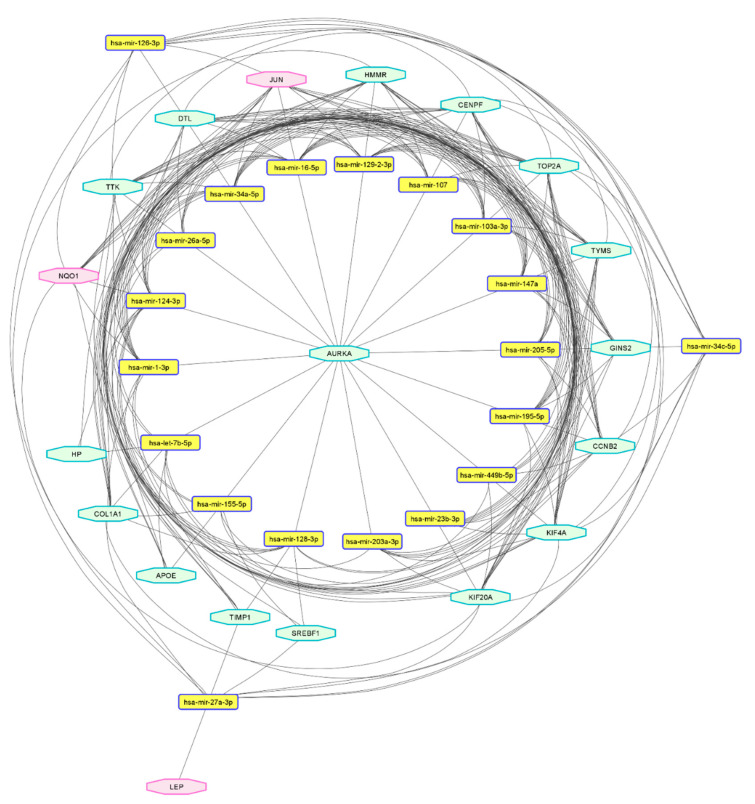
The predicted miRNA-hub gene network shows the interaction between the top 20 miRNA and the hub genes (mRNAs). The hexagons indicate the hub genes, those upregulated marked in the red-purple border and those downregulated in the green border. The round, rectangular nodes are the predicted targeting miRNA. The network contains 19 genes and the top 20 miRNA, with 228 pairs of interactions between them.

**Figure 8 ijms-23-11036-f008:**
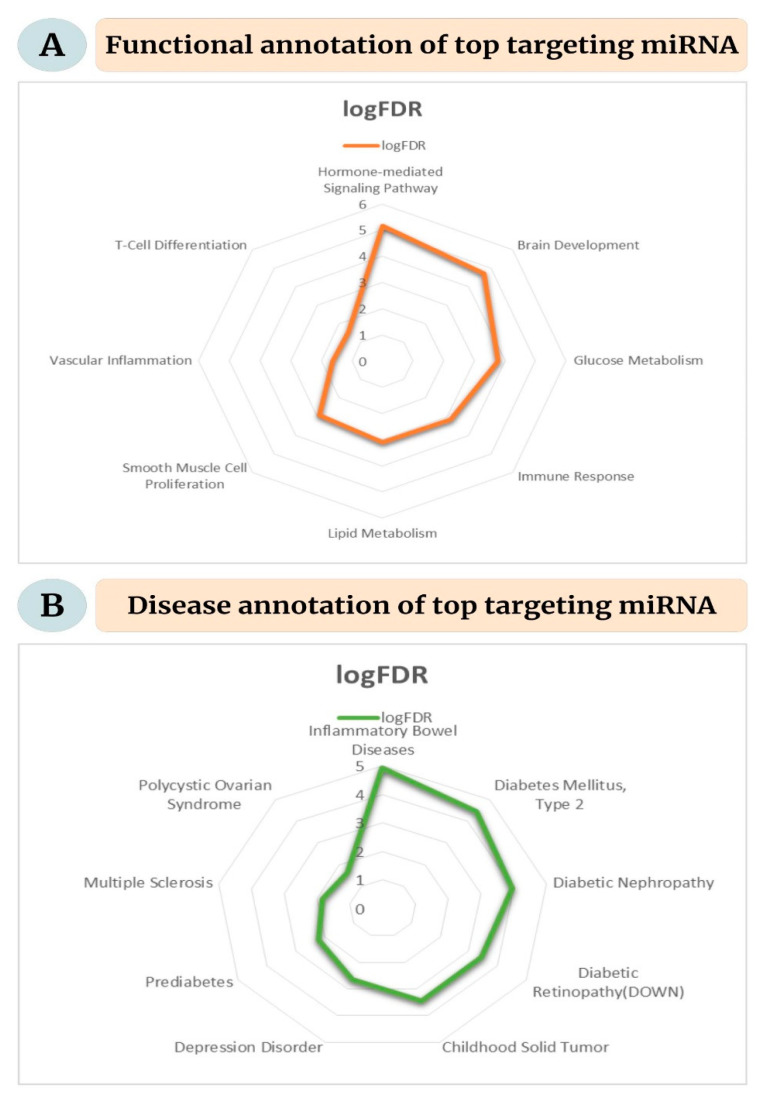
Functional enrichment for the top targeting miRNA. (**A**) Significant functional annotations for the miRNA, (**B**) significant disease associations for the selected miRNA.

**Table 1 ijms-23-11036-t001:** Hub gene analysis with cytoHubba plug-in to identify the top candidate genes.

Sl No.	Official Gene Symbol	Number of Methods Involved
1	TOP2A	10
2	JUN	9
3	APOE	9
4	TIMP1	6
5	COL1A1	5
6	HMMR	5
7	KIF4A	5
8	KIF20A	5
9	TYMS	4
10	LEP	4
11	CENPF	4
12	GINS2	4
13	SREBF1	3
14	HP	3
15	NQO1	3
16	CCNB2	3
17	TTK	3
18	DTL	3
19	AURKA	3

**Table 2 ijms-23-11036-t002:** Candidate hub genes with targeting transcription factors and miRNA.

Candidate Hub Gene	Targeting Transcription Factors	Targeting miRNA
TOP2A	ATF1	**miR-34a-5p**, **miR-16-5p**, **miR-124-3p**, **miR-103a-3p**, **miR-107**, miR-129-2-3p, miR-1-3p, miR-147a, miR-126-3p, miR-27a-3p, miR-195-5p, miR-205-5p, let-7b-5p, miR-26a-5p, miR-23b-3p, miR-128-3p, miR-449b-5p, miR-34c-5p, miR-203a-3p
JUN	CREB1, MYBL2, NFRKB, NRF1, SMAD4, SP3, TFDP1, WT1, ARNT, ZNF382, MEF2D	**miR-34a-5p**, **miR-16-5p**, **miR-124-3p**, **miR-103a-3p**, **miR-107**, miR-129-2-3p, miR-1-3p, miR-147a, miR-126-3p, miR-195-5p, miR-26a-5p, miR-23b-3p, miR-203a-3p, miR-155-5p
APOE	ARNT, ETS1, ATF4, FOXM1, SP1	**miR-34a-5p**, **miR-16-5p**, miR-1-3p, let-7b-5p, miR-155-5p
TIMP1	ARNT, JUND, RELA, SP1, SP3, STAT1	**miR-34a-5p**, **miR-124-3p**, miR-27a-3p, let-7b-5p, miR-26a-5p, miR-128-3p
COL1A1	ATF1, CEBPB, FOXM1, SP1, USF1, WT1, ETS1, MYBL2, RELA, SP1, SP3	**miR-34a-5p**, **miR-16-5p**, **miR-124-3p**, **miR-103a-3p**, **miR-107**, miR-129-2-3p, miR-1-3p, miR-27a-3p, let-7b-5p, miR-128-3p, miR-34c-5p, miR-155-5p
HMMR	ATF1, CREB1, JUNB, MEF2D, NFRKB, SP1	**miR-34a-5p**, **miR-16-5p**, **miR-124-3p**, **miR-103a-3p**, **miR-107**, miR-129-2-3p, miR-1-3p, miR-147a, miR-27a-3p, miR-195-5p, miR-205-5p, miR-23b-3p, let-7b-5p, miR-203a-3p, miR-155-5p
KIF4A	ATF1	**miR-34a-5p**, **miR-16-5p**, **miR-124-3p**, **miR-103a-3p**, **miR-107**, miR-129-2-3p, miR-1-3p, miR-147a, miR-126-3p, miR-27a-3p, miR-195-5p, miR-205-5p, miR-26a-5p, miR-23b-3p, miR-449b-5p, miR-34c-5p, miR-203a-3p
KIF20A	ZNF382	**miR-34a-5p**, **miR-16-5p**, **miR-124-3p**, **miR-103a-3p**, **miR-107**, miR-129-2-3p, miR-147a, miR-126-3p, miR-27a-3p, miR-195-5p, miR-205-5p, miR-23b-3p, miR-449b-5p, miR-34c-5p, miR-203a-3p, miR-155-5p
TYMS	ATF1, CEBPA, NFRKB, SP1, TFDP1, USF1	**miR-34a-5p**, **miR-16-5p**, **miR-103a-3p**, **miR-107**, miR-129-2-3p, miR-1-3p, miR-147a, miR-126-3p, miR-195-5p, let-7b-5p, miR-26a-5p, miR-23b-3p, miR-449b-5p, miR-203a-3p, miR-155-5p
LEP	ATF1, CEBPA, SP1	miR-27a-3p
CENPF	STAT1	**miR-34a-5p**, **miR-16-5p**, **miR-124-3p**, **miR-103a-3p, miR-107**, miR-129-2-3p, miR-1-3p, miR-147a, miR-126-3p, miR-27a-3p, miR-195-5p, miR-205-5p, miR-26a-5p, miR-23b-3p, miR-128-3p, miR-449b-5p, miR-34c-5p
GINS2	-	**miR-34a-5p**, **miR-16-5p**, **miR-124-3p**, **miR-103a-3p, miR-107**, miR-129-2-3p, miR-1-3p, miR-147a, miR-195-5p, miR-205-5p, miR-34c-5p, miR-203a-3p
SREBF1	ATF4, NFRKB, NRF1, RELA, SMAD4, SP3, TFDP1, ZNF382, SP1	**miR-16-5p**, miR-27a-3p, miR-128-3p, miR-155-5p
HP	CEBPB, SMAD4	**miR-124-3p**, miR-147a, let-7b-5p
NQO1	NFRKB, NRF1, JUNB, JUND, NFE2L2	**miR-34a-5p**, **miR-124-3p**, **miR-103a-3p**, **miR-107**, miR-129-2-3p, miR-1-3p, miR-147a, miR-126-3p, miR-27a-3p, miR-205-5p, miR-128-3p
CCNB2	ZNF382, ARNT, NFRKB	**miR-34a-5p**, **miR-16-5p**, **miR-124-3p**, **miR-103a-3p, miR-107**, miR-129-2-3p, miR-147a, miR-195-5p, miR-205-5p, miR-126-3p, miR-23b-3p, let-7b-5p, miR-449b-5p, miR-34c-5p
TTK	-	**miR-34a-5p**, **miR-16-5p**, **miR-124-3p**, **miR-103a-3p**, **miR-107**, miR-129-2-3p, miR-1-3p, miR-147a, miR-126-3p, miR-195-5p, miR-205-5p, miR-26a-5p, miR-128-3p, miR-449b-5p, miR-34c-5p
DTL	JUNB, JUND, MYBL2, NFE2L2, SP1	**miR-34a-5p**, **miR-16-5p**, **miR-124-3p**, **miR-103a-3p**, **miR-107**, miR-129-2-3p, miR-1-3p, miR-147a, miR-126-3p, miR-195-5p, miR-205-5p, miR-26a-5p, miR-128-3p, miR-449b-5p, miR-34c-5p
AURKA	ARNT, NFRKB, ZNF382	**miR-34a-5p**, **miR-16-5p**, **miR-124-3p**, **miR-103a-3p**, **miR-107**, miR-129-2-3p, miR-1-3p, miR-147a, miR-195-5p, miR-205-5p, miR-26a-5p, miR-23b-3p, let-7b-5p, miR-128-3p, miR-449b-5p, miR-203a-3p, miR-155-5p

The top five targeting miRNA for each candidate hub gene are marked in bold font. GINS2 and TTK had no common transcription factor between the two databases—TTRUST and ENCODE.

## Data Availability

Publicly available Gene Expression Omnibus datasets were used for the analysis in this study. These datasets can be accessed at: (1) https://www.ncbi.nlm.nih.gov/geo/query/acc.cgi?acc=GSE9624 (GSE9624; accessed on 11 July 2022), and (2) https://www.ncbi.nlm.nih.gov/geo/query/acc.cgi?acc=GSE88837 (GSE88837; accessed on 11 July 2022).

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
