# Peer review of "Visceral Adipose Tissue Molecular Networks and Regulatory microRNA in Pediatric Obesity: An In Silico Approach"

_ijms, 2022, doi:10.3390/ijms231911036_

Round 1

Reviewer 1 Report

This is a very well written and thorough study on the potential mechanisms for the development of obesity and obesity related complications in children(2-9 y/o). The scientific analysis is sound and the conclusions are well supported by the data.

Author Response

Reviewer #1: This is a very well written and thorough study on the potential mechanisms for the development of obesity and obesity related complications in children (2-19 y/o). The scientific analysis is sound and the conclusions are well supported by the data.

Ans. The authors appreciate the kind remarks of the reviewer.

Reviewer 2 Report

General comment

The paper studies the gene expression pattern and its regulatory network in the visceral adipose tissue in obese children. The topic of the manuscript is of very interesting, since the prevalence of obese nutritional status in children is increasing in the societies and the genetic background of this metabolic syndrome has not been explored yet. Although the sample size in the two datasets are very low, I recommend to accept this paper for publication in International Journal of Molecular Sciences after a minor revision because of its originality. The main reason: the figures are incomprehensible (the size of the legends are too small, the relations among the studied genes are also invisible, the captions are too short and cannot be interpreted without the whole manuscript, the readers cannot understand the figures).

I list my other specific/minor comments and suggestions to the manuscript in the order of the chapters of the manuscript to help the revision:

Introduction

I1: “In the past few decades, obese children and adolescents have sharply increased in 37 developing and developed countries.” – please correct this sentence, I guess the number of obese children has increased.

Figures

F1: The caption of Figure 2 should contain more details, not only down and up for colours, by explaining how obese vs. lean children are compared in the calculations.

F2: Please increase the size of the letters in Figure 2

F3: Capture in Figures 4-5-6-7 is absolutely invisible, please somehow change the letter size.

F4: Please give statistics to Figure 5 with the clusters, only Cluster 1 got information on its reliability, it cannot be evaluated how the other 5 clusters support the statement.

Discussion

D1: A summary table with the names of the genes that were found to be connected in this regulatory network in the present analyis should help the summary of the results.

Author Response

Reviewer #2: General comment

The paper studies the gene expression pattern and its regulatory network in the visceral adipose tissue in obese children. The topic of the manuscript is of very interesting, since the prevalence of obese nutritional status in children is increasing in the societies and the genetic background of this metabolic syndrome has not been explored yet. Although the sample size in the two datasets are very low, I recommend to accept this paper for publication in International Journal of Molecular Sciences after a minor revision because of its originality. The main reason: the figures are incomprehensible (the size of the legends are too small, the relations among the studied genes are also invisible, the captions are too short and cannot be interpreted without the whole manuscript, the readers cannot understand the figures).

Ans. The authors appreciate the kind remarks of the reviewer. According to the suggestion, we have remade the figures to make the letters comprehensible to the readers. We have also revised the captions to be interpreted without the whole manuscript.

I list my other specific/minor comments and suggestions to the manuscript in the order of the chapters of the manuscript to help the revision:

Introduction

I1: “In the past few decades, obese children and adolescents have sharply increased in 37 developing and developed countries.” – please correct this sentence, I guess the number of obese children has increased.

Ans. We had not used 37 (or any specific number) as the number of countries where the prevalence of obese children and adolescents has increased, as we could not find such information in the literature. The sentence in the revised version reads – “In the past few decades, the prevalence of obesity in the pediatric age group has sharply increased in both developing and developed countries.”

Figures

F1: The caption of Figure 2 should contain more details, not only down and up for colours, by explaining how obese vs. lean children are compared in the calculations.

F2: Please increase the size of the letters in Figure 2

Ans. We thank the reviewer for these constructive comments. We have now revised the caption of Figure 2 and included details regarding how obese vs. lean children are compared in the calculations. We have also increased the font size and restructured the entire figure 2.

F3: Capture in Figures 4-5-6-7 is absolutely invisible. Please somehow change the letter size.

F4: Please give statistics to Figure 5 with the clusters, only Cluster 1 got information on its reliability, it cannot be evaluated how the other 5 clusters support the statement.

Ans. We have re-created figures 4, 5, 6, 7. We hope that in the revised renditions, the letters will be visible. Regarding clusters, the nodes, edges, and scores for all the clusters have been added in text, the figure legend (Figure 5), and the cut-offs used for the cluster analysis.

Discussion

D1: A summary table with the names of the genes that were found to be connected in this regulatory network in the present analyis should help the summary of the results.

Ans. The authors appreciate the suggestion from the reviewer. We have added a summary table (Table 2) with the name of the genes taking part in the regulatory network and the targeting transcription factors and miRNA. We have also revised and expanded the discussion section to summarize our findings better.

Reviewer 3 Report

Comments: Pathological expansion of visceral adipose tissue in obesity is often associated with metabolic disorders and insulin resistance in both children and adults. Identifying factors that lead to visceral adipose dysfunction could inform future therapy for obesity-associated metabolic disease. The authors in the manuscript used publicly available datasets to examine transcriptomics and gene regulatory networks in obese pediatric individuals. The authors use comprehensive data analysis tools to investigate networks, pathways, genes, etc. changes in obese individuals. However, there are several concerns that limit the enthusiasm for a broad readership.

1)    The paper mostly reads as a report identifying up and down-regulated features of the dataset and lacks a unifying theme that could inform something previously unknown or provide a basis for future investigation.

2)    The texts in the figures are barely legible and it is unclear how the transcriptional network changes in the dataset relate to disease progression and state.

3)    The study ultimately focuses on miRNA and there are no contexts of how these miRNAs are relevant to metabolic disorders.

4)    Are some of the transcription factors and networks involved in miRNA gene regulation?

5)    Focusing on the miRNAs and how these miRNAs cause changes in visceral adipogenesis by experiment would have been more informative.

Author Response

Reviewer #3: Comments: Pathological expansion of visceral adipose tissue in obesity is often associated with metabolic disorders and insulin resistance in both children and adults. Identifying factors that lead to visceral adipose dysfunction could inform future therapy for obesity-associated metabolic disease. The authors in the manuscript used publicly available datasets to examine transcriptomics and gene regulatory networks in obese pediatric individuals. The authors use comprehensive data analysis tools to investigate networks, pathways, genes, etc. changes in obese individuals. However, there are several concerns that limit the enthusiasm for a broad readership.

1)    The paper mostly reads as a report identifying up and down-regulated features of the dataset and lacks a unifying theme that could inform something previously unknown or provide a basis for future investigation.

Ans. The authors thank the reviewer for this suggestion. We have revised our discussion section within the context of prevalent themes in our findings, including the TGFβ signaling pathway, immune dysfunction, and apoptosis. We have further added relevant material from the literature discussing our findings.

Abstract

Page 14: Discussion – paragraphs 2 and 3

2)    The texts in the figures are barely legible and it is unclear how the transcriptional network changes in the dataset relate to disease progression and state.

Ans. We have revised the figures and hope that the networks will be legible in the new figures. The role of the transcription factors within the context of the hub genes and gene ontology has been elaborated in the “Discussion” section.

Figures 1-7

Table 2 for network summary

Page 14-15: Discussion – Para 4 and 5

3)    The study ultimately focuses on miRNA and there are no contexts of how these miRNAs are relevant to metabolic disorders.

4)    Are some of the transcription factors and networks involved in miRNA gene regulation?

Ans. The authors thank the reviewer for the valuable suggestions. The miRNA is indeed one of the focuses of this study. We have, therefore, expanded the section on miRNA, 1) adding a contextual part in the “Introduction”, and 2) elaborating on our findings in the “Discussion”. We have added the “regulatory miRNA” in the title.

We have also included material from the literature on the transcription factors and hub genes from the network involved in miRNA regulation.

Page 1: Title

Page 2: Introduction – Para 4

Page 15: Discussion – Para 5 and 6

5)    Focusing on the miRNAs and how these miRNAs cause changes in visceral adipogenesis by experiment would have been more informative.

Ans. The authors agree that experimental validation of the miRNA in visceral adipogenesis would have been more informative. However, this study lacks any funding sources. Hence we had to take an entirely in silico approach.

We look forward to validating our findings, provided we obtain the necessary funds in the future, but as of now, we have to go forward with our current findings. Hopefully, the reviewer will be kind enough to consider our predicament.

Round 2

Reviewer 3 Report

Comments addressed.